# G-quadruplex DNA drives genomic instability and represents a targetable molecular abnormality in ATRX-deficient malignant glioma

Yuxiang Wang[1], Jie Yang[2], Aaron T. Wild[1], Wei H. Wu[1], Rachna Shah [1], Carla Danussi[3], Gregory J. Riggins[4], Kasthuri Kannan[5], Erik P. Sulman [2,6], Timothy A. Chan [1,7] & Jason T. Huse[3,6]

Mutational inactivation of *ATRX* (α-thalassemia mental retardation X-linked) represents a defining molecular alteration in large subsets of malignant glioma. Yet the pathogenic consequences of ATRX deficiency remain unclear, as do tractable mechanisms for its therapeutic targeting. Here we report that ATRX loss in isogenic glioma model systems induces replication stress and DNA damage by way of G-quadruplex (G4) DNA secondary structure. Moreover, these effects are associated with the acquisition of disease-relevant copy number alterations over time. We then demonstrate, both in vitro and in vivo, that ATRX deficiency selectively enhances DNA damage and cell death following chemical G4 stabilization. Finally, we show that G4 stabilization synergizes with other DNA-damaging therapies, including ionizing radiation, in the ATRX-deficient context. Our findings reveal novel pathogenic mechanisms driven by ATRX deficiency in glioma, while also pointing to tangible strategies for drug development.

[1] Human Oncology and Pathogenesis Program, Memorial Sloan-Kettering Cancer Center, New York, NY 10065, USA. [2] Department of Radation Oncology, University of Texas MD Anderson Cancer Center, Houston, TX 77030, USA. [3] Department of Translational Molecular Pathology, University of Texas MD Anderson Cancer Center, Houston, TX 77030, USA. [4] Departments of Neurosurgery, Oncology, and Genetic Medicine, Johns Hopkins School of Medicine, Baltimore, MD 21231, USA. [5] Department of Pathology, New York University School of Medicine, New York, NY 10016, USA. [6] Department of Pathology, University of Texas MD Anderson Cancer Center, Houston, TX 77030, USA. [7] Department of Radiation Oncology, Memorial Sloan-Kettering Cancer Center, New York, NY 10065, USA. Correspondence and requests for materials should be addressed to J.T.H. (email: jhuse@mdanderson.org)

nfiltrating gliomas are the most common primary brain tumors and, despite considerable molecular and clinical heterogeneity, remain uniformly deadly in the face of aggressive surgical and cytotoxic treatment regimens[1]. Recent large-scale genomic profiling has shown that inactivating mutations in *ATRX* (α-thalassemia mental retardation X-linked) characterize large subclasses of both adult and pediatric glioma[2–4]. Despite these striking correlations, however, the precise mechanisms by which *ATRX* mutation promotes gliomagenesis remain unclear. Recent reports have linked germline *ATRX* mutations to osteosarcoma[5–7], and their association with a rare, congenital neurodevelopmental condition (ATR-X syndrome) is well-established[8]. *ATRX* encodes a chromatin binding protein widely implicated in epigenetic regulation and remodeling[9–15], suggesting that epigenomic dysfunction may, at least in part, underlie the oncogenic effects of ATRX deficiency. *ATRX* loss has also been implicated in alternative lengthening of telomeres (ALT), an abnormal telomerase-independent mechanism of telomere maintenance based on homologous recombination[16,17]. Finally, ATRX deficiency has been repeatedly linked to replication stress, DNA damage, copy number alterations (CNAs), and aneuploidy[18–23], and recent work has associated ATRX deficiency specifically with copy number loss at ribosomal DNA loci[24]. Whether and how such genomic instability contributes to the initiation and/or evolution of malignant glioma remains unclear.

ATRX binds widely across the genome at sites featuring tandem repeats and CpG islands[25]. Many such loci are GC-rich and susceptible to forming G-quadruplexes (G4s), abnormal secondary structures implicated in both transcriptional dysregulation and DNA damage. Accordingly, it has been hypothesized that, among its various functionalities, ATRX serves to resolve G4s genome-wide and mitigate their deleterious consequences[25,26]. The tendency of G4s to stall replication forks underlies their association with DNA damage[27]. Chemical stabilization of G4s induces replication stress at genomic loci prone to G4 formation[28], and also promotes DNA damage and apoptosis in neural progenitor cells[29]. Moreover, recent work suggests that G4-induced replication stress at telomeres may drive ALT in the ATRX-deficient setting through induction of homologous recombination[16]. Indeed, G4 stabilization hampers the ability of forced ATRX expression to abrogate the ALT phenotype in vitro. Taken together, these findings provide compelling links between ATRX, G4 biology, and genomic instability. Whether ATRX deficiency directly induces G4 formation and DNA damage, however, remains unestablished, as does the impact of G4s on the pathogenesis of ATRX-deficient neoplasia. Moreover, therapeutic strategies leveraging G4 biology in the selective targeting of ATRX-deficient cancers have not been extensively explored.

To characterize the role of G4-mediated genomic instability in glioma biology, we inactivated ATRX in isogenic normal human astrocyte (NHA) and glioma stem cell (GSC) models. We found that ATRX loss increased G4 formation, replication stress, and DNA damage genome-wide. Moreover, ATRX-deficient NHAs accumulated clinically relevant CNAs at an accelerated rate relative to ATRX-intact counterparts. Chemical G4 stabilization was associated with enhanced DNA damage and cell death in ATRX-deficient contexts. Moreover, ATRX-mutant GSC xenografts were selectively sensitive to G4-targeting in vivo. Finally, G4 stabilization in ATRX-deficient NHAs and GSCs effectively synergized with other DNA-damaging treatment strategies, including ionizing radiation. These findings clarify distinct mechanisms by which G4s influence ATRX-deficient glioma pathogenesis and indicate that G4 stabilization may represent an attractive therapeutic strategy for the selective targeting of ATRX-mutant cancers.

## Results

**ATRX deficiency promotes G4 formation and DNA damage.** To model the genomic consequences of ATRX deficiency in a glioma-relevant cellular context, we performed shRNA-mediated ATRX knockdown in TERT and E6/E7-transformed NHAs. Several studies have effectively employed immortalized NHAs to delineate key aspects of glioma biology[30–34]. In our investigations, we employed two distinct hairpin constructs to silence *ATRX*—shATRX1 and shATRX2—the latter of which was driven by a doxycycline (dox)-inducible promoter (Fig. 1a). This framework allowed for the analysis of both immediate and long-term effects of ATRX deficiency as well as their reversibility. Using a monoclonal antibody known to recognize G4 structures in situ (1H6), we then demonstrated that ATRX deficiency increased nuclear G4s relative to levels seen in control shRNA-expressing parental NHAs (shCon), an effect that was reversible upon restored ATRX expression (Fig. 1b–c). Increased G4s were also found in p53-deficient murine neuroepithelial progenitor cells (mNPCs) featuring inactivated *Atrx* (Supplementary Fig. 1a).

The specificity of 1H6 for DNA-based secondary structures was confirmed by DNAase treatment, which eradicated immunolabelling, and RNase treatment, which did not, in NHAs treated with the G4-stabilizing agent CX-3543 (see below, Supplementary Fig. 1b). Moreover, the effects of ATRX knockdown on nuclear G4 levels in isogenic NHAs, as assessed by 1H6 immunofluorescence, were recapitulated with a different G4-targeting monoclonal antibody (BG4; Supplementary Fig. 1c). Forced expression of the isocitrate dehydrogenase 1 (IDH1) R132H mutation in our isogenic NHAs did not significantly alter G4 levels (as assessed by BG4) in either the ATRX-intact or the ATRX-deficient context (Supplementary Fig. 1d). ATRX deficiency almost invariably co-occurs with mutations in *IDH1* or its homologue *IDH2* in adult gliomas. We also compared GSCs derived from IDH-mutant, *ATRX*-mutant (08-0537) and IDH-mutant, *ATRX* wild-type (TS 603) gliomas, finding increased nuclear G4s in the former by BG4 immunofluorescence (Fig. 1d). Finally, we found that ATRX knockdown (sh590) in a glioblastoma-derived GSC (TS 543; IDH and *ATRX* wild type) enhanced G4 formation (Fig. 1e and Supplementary Fig. 1e). These data show that, in multiple glioma-relevant cellular contexts, ATRX deficiency promotes G4 accumulation.

To further support these findings, we employed a synthetic single-chain antibody (hf2) to immunoprecipitate G4s in both ATRX-intact and ATRX-deficient contexts. hf2 specificity was validated by gel-shift assay showing specific capture of synthesized Kit2 nucleotides independently from random ssDNA and dsDNA (Supplementary Fig. 2a). We then performed pulldowns in our isogenic NHAs, finding that ATRX deficiency significantly increased the qPCR enrichment of known G4 sites within the *MYC* and *ZNF618* loci, as well as in telomeric regions on chromosomes 1, 2, and X (Fig. 1f)[35–37]. Consistent with the notion that ATRX resolves G4s as part of its normal functionality, we found a distinct absence of colocalization between ATRX and G4 immunofluorescence in ATRX-intact NHAs (Fig. 1g). Finally, functional studies demonstrated that ATRX knockdown failed to induce significant changes in apoptosis, BrdU incorporation, or cell cycle profile (Supplementary Fig. 2b–d). Taken together, these findings confirm, in a true isogenic system, that ATRX deficiency promotes G4 formation. Moreover, they indicate that, at least in this glioma-relevant context, increased G4s as a consequence of ATRX deficiency are insufficient to drive apoptosis or impact cellular proliferation.

We then examined whether the G4s induced by ATRX deficiency promoted replication stress and DNA damage, as suggested by prior literature[38]. We found that ATRX knockdown significantly and reversibly increased γ-H2AX and

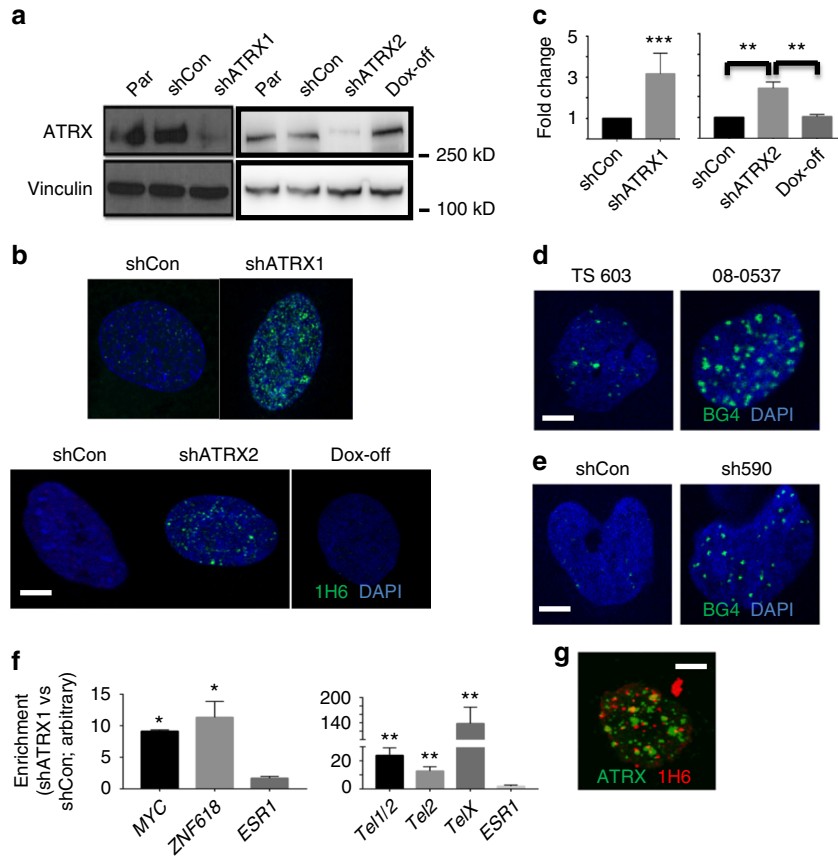

**Fig. 1** ATRX deficiency promotes G4 formation. **a** Western blots for ATRX in parental (Par), shControl (shCon), and shATRX NHA (Vinculin loading control). Left panel shows constitutive NHA lines (shATRX1) and right panel shows the inducible lines post Doxycycline induction (shATRX2) and withdrawal (Dox-off). **b** Immunofluorescent staining of G4 (1H6) in constitutive (upper panel) and inducible (lower panel) NHA lines (DAPI counterstain). **c** Quantified relative G4 signal intensity (50 nuclei counted in all cases). **d**, **e** Immunofluorescent staining of G4 (BG4) in *ATRX*-intact (TS 603) and *ATRX*-mutant (08-0537) GSCs (**d**) and in TS 543 GSCs (**e**) subjected to ATRX knockdown (sh590) versus control (shCon; DAPI counterstain). **f** G4-containing DNA fragments from shCon and shATRX1 NHAs (three replicates each) were pulled down from sheared genomic DNA with purified hf2 antibodies. Recovered DNA was subjected to real-time PCR for telomeric sequences or G4-rich promoter regions of *MYC and ZNF618* (*ESR1* used as negative control). Graphs show enrichment over GAPDH scaled to shCon levels. **g** ATRX and G4 immunofluorescence in parental NHAs showing no significant colocalization. Where applicable, error bars reflect SEM; *P* values determined by unpaired, two-tailed *t*-test (\**P* < 0.05, \*\**P* < 0.01, \*\*\**P* < 0.001); scale bars represent 10 µm

53BP1-positive DNA damage foci by immunofluorescence (Fig. 2a–c), and did so in a pattern that extensively colocalized with nuclear G4 distribution, as assessed by both 1H6 and BG4 (Fig. 2c and Supplementary Fig. 3). Moreover, these changes were accompanied by engagement of the replication stress pathway, as evidenced by upregulated levels p-CHK1 and p-KAP1 on western blot (Fig. 2d). To further ascertain the extent of colocalization between G4s and DNA damage sites in the ATRX-deficient context, we performed chromatin immunoprecipitation (ChIP) for γ-H2AX and Bloom syndrome RecQ-like helicase (BLM), a protein known to bind G4 DNA[39]. We found that ATRX loss significantly increased both γ-H2AX and BLM enrichment at putative G4 sites within the *MYC* and *ZNF618* loci (Fig. 2e), far exceeding effects at a negative control locus (*ESR1*). To determine whether increased levels of DNA damage in the ATRX-deficient setting might lead to structural abnormalities in chromosomes, we performed metaphase cytogenetic analysis coupled with telomere-fluorescence in situ hybridization (FISH) in shCon and shATRX1 NHAs uniformly aged to 15 passages. Consistent with multiple prior reports, ATRX knockdown in this context was not associated with an obvious ALT phenotype (Supplementary Fig. 4a, b). However, we consistently observed

higher levels of chromosome breakage in ATRX-deficient NHAs relative to shCon counterparts (Fig. 3a and Supplementary Fig. 4b). Increased chromosome breaks were also observed in *ATRX*-mutant 08-0537 GSCs relative to *ATRX* wild-type TS 603 GSCs (Fig. 3b and Supplementary Fig. 4c), and a similar trend, though not statistically significant, was seen in p53-deficient mNPCs also featuring *Atrx* inactivation (versus p53-deficient, *Atrx*-intact isogenics; Fig. 3c and Supplementary Fig. 4d). These data establish pathogenic links between G4s arising with ATRX deficiency and the generalized genomic instability characteristic of ATRX-mutant tumors and cell lines.

**ATRX deficiency drives clinically relevant CNA formation.** Having confirmed that ATRX deficiency induces DNA damage and structural abnormalities in chromosomes, likely through G4-mediated mechanisms, we sought to assess whether these biological processes might promote acquisition of CNAs in ATRX-deficient tumors. *ATRX* mutations in adult glioma arise almost exclusively in the setting of concurrent mutations in *TP53* and either *IDH1* or *IDH2*. The glioma subtype defined by this combined genotype, termed "IDHmut-noncodel"[2], features

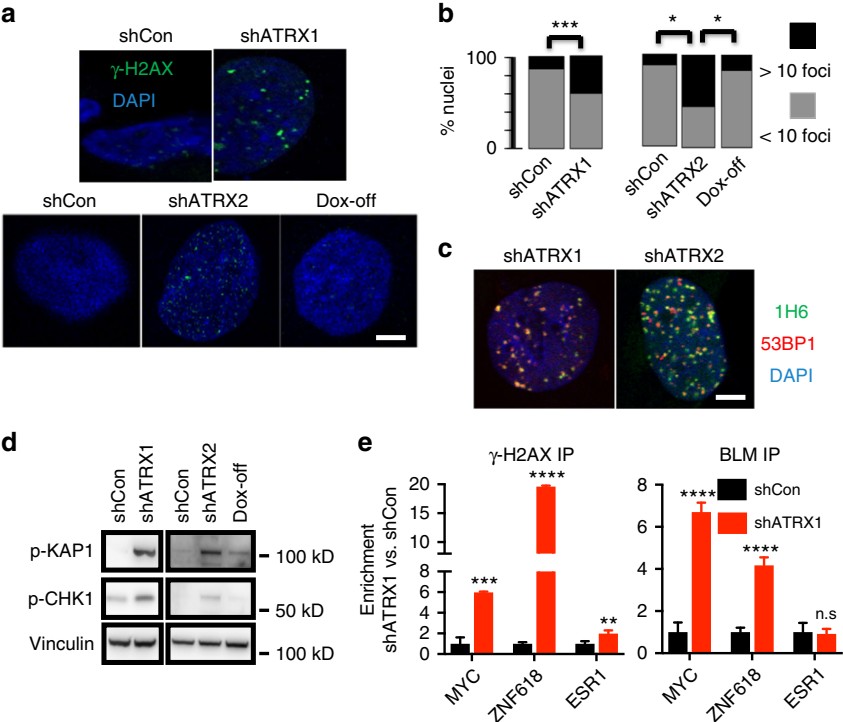

**Fig. 2** ATRX deficiency promotes replication stress and DNA damage. **a** γ-H2AX immunofluorescence in constitutive (upper panel) and inducible (lower panel) NHAs (DAPI counterstain). **b** Quantified percentage of cells with >10 immunopositive γ-H2AX foci (50 nuclei counted in all cases). **c** NHAs with ATRX knockdown were double stained for G4 and 53BP1, showing extensive colocalization. **d** Western blots of p-KAP1 and p-CHK1 show activation of replication stress arising with ATRX knockdown. **e** γ-H2AX and BLM ChIP in shCon and shATRX1 NHAs (three replicates each) showing that ATRX deficiency promotes enrichment for both proteins at putative G4-rich promoter regions (MYC and ZNF618) far exceeding that seen at a negative control site (ESR1). Where applicable, error bars reflect SEM; $P$ values determined by unpaired, two-tailed $t$-test (*$P < 0.05$, **$P < 0.01$, ***$P < 0.001$, ****$P < 0.0001$, n.s.: $P > 0.05$); scale bars represent 10 μm

almost uniformly low-level *ATRX* expression and exhibits a characteristic pattern of CNAs, distinct form that commonly seen in other adult glioma subtypes (Supplementary Fig. 5)[2]. More-over, multiple CNAs recurrently featured in ATRX-deficient glioma mobilize established oncogenic and/or tumor suppressive loci, including *MYC* and *CDKN2A*[2], implying that such structural abnormalities may contribute to the malignant evolution of this inexorably progressive cancer.

To experimentally model CNA formation in the ATRX-deficient setting, we aged our isogenic NHAs in culture, monitoring DNA copy number by SNP array at 5 and 15 passages—~1 month and ~3 months in culture, respectively (Supplementary Fig. 5). We found that while both sets of isogenics demonstrated increased CNAs over time, ATRX-deficient NHAs exhibited a distinct pattern of gains and losses that included larger (>1 Mb) alterations not seen in ATRX-intact counterparts (Fig. 4a). Analysis of TCGA SNP data revealed a similar subset of broad alterations included within the CNA profile of the IDHmut-noncodel glioma subtype (Fig. 4b). Moreover, two of the broad CNAs arising with ATRX deficiency in NHAs, involving 12p gain and 14q loss, recapitulated events commonly seen in the IDHmut-noncodel glioma subtype and associated with relatively unfavorable prognosis when present (Fig. 4c–f). Taken together, these findings support the notion that G4-mediated DNA damage induces specific patterns of CNAs in the ATRX-deficient, IDHmut-noncodel glioma subtype, which in turn influence malignant evolution.

**Chemical G4 stabilization selectively targets ATRX-deficient cells.** As indicated above, the pronounced effects of ATRX

deficiency on G4 formation and replication stress in NHAs were not associated with increased cell death at baseline. Nevertheless, we reasoned that compensatory mechanisms to resolve G4s and otherwise maintain genomic integrity were likely under increased stress, and that chemical stabilization of G4s might, therefore, selectively enhance DNA damage to an unsustainable degree in the ATRX-deficient context. To evaluate the therapeutic potential of this synthetic lethal approach, we treated our isogenic NHAs in culture with increasing concentrations of CX-3543 (Quarfloxin), an established G4-stabilizing agent[40,41]. We found that ATRX knockdown, in both constitutive and inducible systems, was associated with increased sensitivity to CX-3543 (IC50 = 42.449 nM (shATRX1) vs 328.835 nM (shCon) in constitutive NHAs and IC50 = 357.424 nM (pre-induction) vs 53.415 nM (shATRX2) vs 247.700 nM (post-induction) in inducible NHAs; Fig. 5a, b). Similar results were obtained with two other G4-stabilizing agents, pyridostatin (PDS) and CX-5461 (Supplementary Fig. 6a–b)[28,42]. Clonogenicity studies also revealed enhanced vulnerability to CX-3543 in ATRX-deficient NHAs as well as TS 543 GSCs subjected to ATRX knockdown (sh590; Fig. 5c, d). Restoring ATRX expression reverted NHAs to baseline levels of sensitivity (Fig. 5b).

γ-H2AX immunofluorescence demonstrated dramatically increased levels of DNA damage in ATRX-deficient NHAs treated with CX-3543, accompanied by activation of the replication stress pathway as determined by western blot (Fig. 5e–h). Similar effects on γ-H2AX immunofluorescence were observed using either PDS or CX-5461 in NHAs (Supplementary Fig. 7a), and were also seen in both ATRX-knockdown TS 543 and *ATRX*-mutant 08-0537 GSCs treated with CX-3543, each

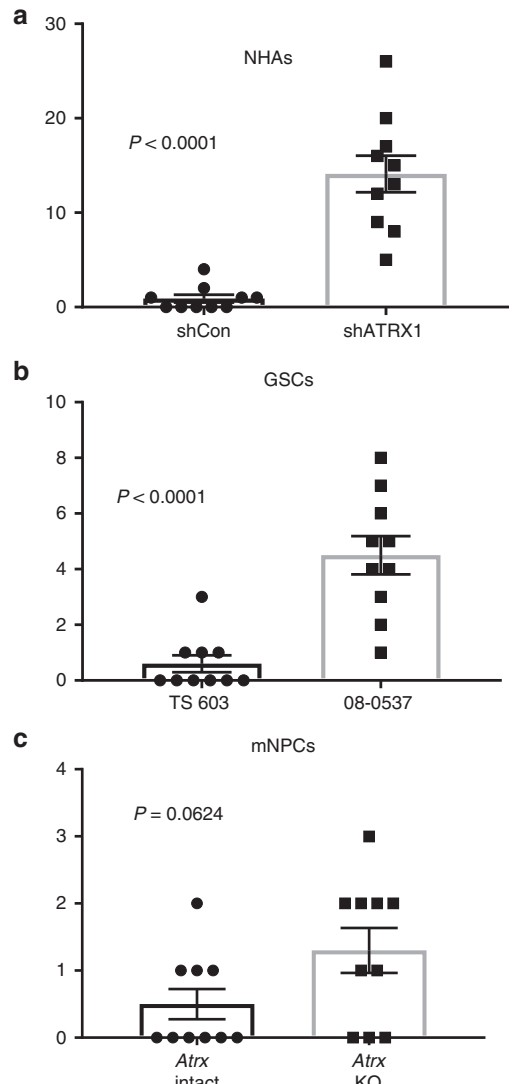

**Fig. 3** ATRX deficiency induces chromosome breaks. **a** ATRX-deficient NHAs (shATRX1, passage 15) showed significantly increased chromosome breaks by cytogenetic analysis relative to ATRX-intact controls (shCon). **b** *ATRX*-mutant GSCs (08-0537) showed significantly increased chromosome breaks by cytogenetic analysis relative to *ATRX* wild-type GSCs (TS 603). **c** Quantified chromosome breaks in *Atrx*-intact and *Atrx*-KO mNPCs (also *Tp53*-/-, passage 10). In all cases, 10 sets of chromosomes were quantified. Error bars reflect SEM; *P* values determined by unpaired, two-tailed *t*-test

relative to ATRX-intact counterparts (Supplementary Fig. 7b). Moreover, 53BP1-positive DNA damage foci arising with CX-3543 demonstrated extensive colocalization with G4s on confocal microscopy (Supplementary Fig. 8a–b), a finding recapitulated by γ-H2AX immunofluorescence in ATRX-deficient NHAs treated with either CX-3543, PDS, or CX-5461, using either BLM or BG4 immunofluorescence to designate G4s (Supplementary Fig. 8c–d). Once again, these effects were reversed following ATRX re-expression (Fig. 5f, h, and Supplementary Fig. 8b). Finally, G4 stabilization in ATRX-deficient NHAs bolstered the extent of ChIP enrichment for both BLM and γ-H2AX at putative G4 sites (Supplementary Fig. 9a–9b), further demonstrating that causal links between impaired G4 resolution and DNA damage underlie our synthetic lethal approach. Annexin V flow-cytometry confirmed that the heightened sensitivity of

ATRX-deficient NHAs to CX-3543 reflected increased apoptosis, and this enhanced cell death followed the kinetics of replication stress pathway activation in both constitutive and inducible isogenic contexts (Fig. 6a–d). Analogous experiments in ATRX-intact and ATRX-knockdown TS 543 GSCs yielded similar findings (Supplementary Fig. 10a–b). In total, these results indicate that chemical stabilization of G4 structures selectively promotes cell death in the ATRX-deficient context, likely by inducing toxic levels of DNA damage.

The experimental links, described above, between replication stress, DNA damage, and CX-3543 treatment prompted us to consider whether G4 stabilization might enhance the therapeutic efficacy of established DNA-damaging treatment strategies, particularly in ATRX-deficient context. To evaluate this possibility, we subjected vehicle and CX-3543-treated isogenic NHAs, cultured in soft agar, to increasing doses of either ionizing radiation (IR) or hydroxyurea (HU), assessing viable colonies at 21 days. We found that CX-3543 treatment potentiated the cytotoxicity of both IR and HU, and while these effects were significant for both NHA genotypes, they were particularly strong in the setting of ATRX deficiency (Fig. 6e, f). Restoring ATRX expression markedly dampened the extent of cytotoxic synergy (Fig. 6e, f). Moreover, the ATRX-dependent radiosensitization properties of CX-3543 were recapitulated in TS 543 GSC isogenics, (Supplementary Fig. 10c). These findings inform additional therapeutically relevant strategies combining chemical G4 stabilization with well-established treatment modalities in the targeting of ATRX-deficient cancer.

**ATRX loss enhances sensitivity to G4 stabilization in vivo.** Having established the increased sensitivity of ATRX-deficient NHAs and GSCs to chemical G4 stabilization in cell culture, we sought to ascertain whether this approach could exhibit a similar degree of efficacy in vivo. To this end, we employed an *ATRX*-mutant, patient-derived GSC line (JHH-273) capable of forming tumors in murine hosts when embedded in the hind flank[43]. Following cellular implantation, we subjected xenografted mice to daily intravenous treatment with either CX-3543 or vehicle and monitored tumor growth over time. We found that CX-3543 dramatically slowed the growth of JHH-273 flank tumors (Fig. 7a, b and Supplementary Fig. 11a) and significantly prolonged survival in xenografted mice (Fig. 7c). Histopathological examination of CX-3543-treated xenografts revealed cellular depopulation, reduced proliferative activity by Ki-67 immunostaining, and increased γ-H2AX-positive DNA damage foci relative to untreated counterparts, recapitulating in vitro findings (Fig. 7d). Notably, telomere FISH failed to reveal changes in the level of ALT in residual viable tumor following CX-3543 treatment (Fig. 7d).

To ascertain whether these effects were specific to the *ATRX*-mutant context, we performed analogous xenograft experiments using *ATRX* wild-type TS 543 cells. In these studies, we found that CX-3543 treatment had no appreciable effect on either xenograft growth or mouse survival (Fig. 8a, b and Supplementary Fig. 11b). However, when we subjected these same GSCs to ATRX knockdown, they were rendered sensitive to CX-3543 to an extent similar to that observed for JHH-273 cells (Fig. 8c, d and Supplementary Fig. 11c). ATRX knockdown also recapitulated histopathological effects on cellular depopulation, proliferative activity, and γ-H2AX-positivity (Fig. 8e, f). Consistent with prior reports[19,22], ATRX knockdown was not associated with ALT in TS 543 cells (Fig. 8e, f). Taken together, these in vivo findings further support the therapeutic potential for chemical G4 stabilization in the selective targeting of ATRX-deficient glioma.

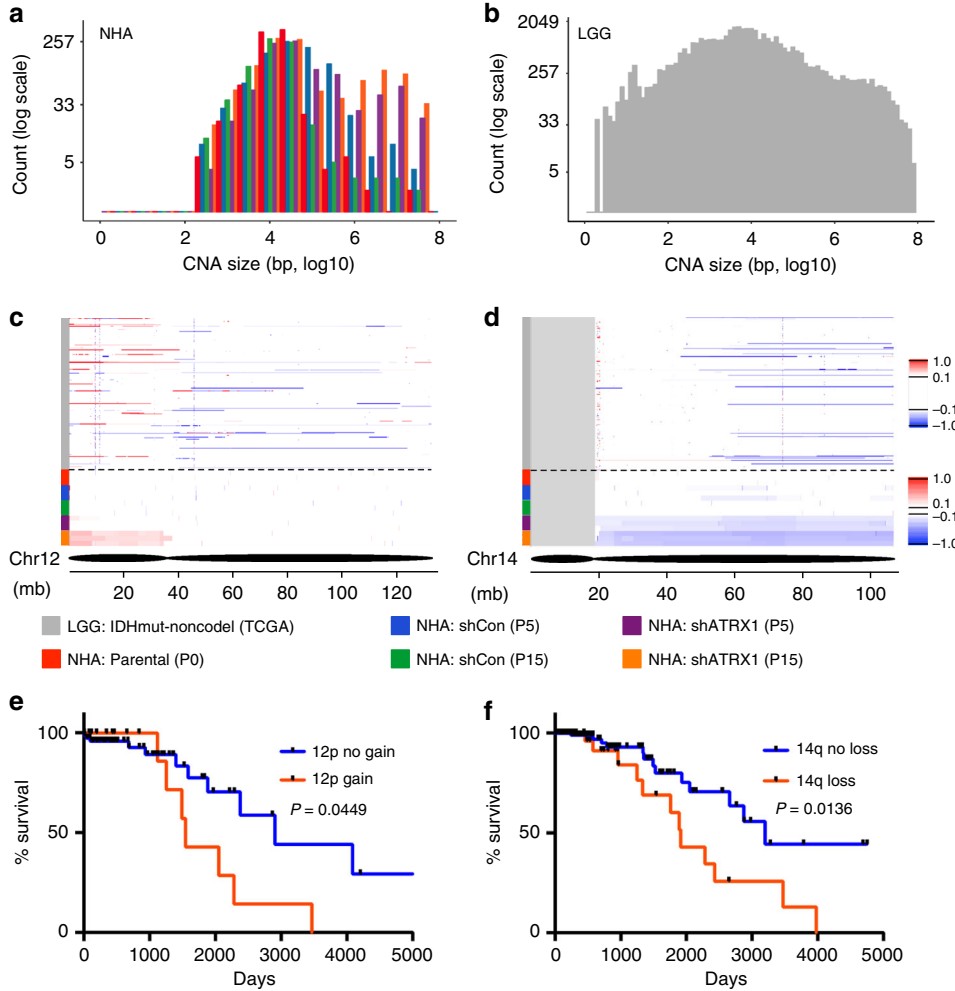

**Fig. 4** ATRX deficiency induces clinically relevant copy number alterations. **a** Size distribution of CNAs in parental (P0), shCon (P5 and P15), and shATRX1 (P5 and P15) NHAs (three replicates each). In ATRX knockdown cells, large CNAs (>1 Mb) arise over passage number. **b** Size distribution of CNAs in IDHmut-noncodel gliomas (TCGA)[2]. **c**, **d** IGV plots for chromosome 12 (**c**) and 14 (**d**) comparing CNA regions in IDHmut-noncodel gliomas (above dotted line) to NHAs (below dotted line). Color scales indicate log₂ copy number. **e**, **f** Kaplan–Meier curves for IDHmut-noncodel glioma patients with or without either 12p gain or 14q loss, showing significant differences in overall survival. $P$ values determined by Log-rank (Mantel–Cox) test

## Discussion

As indicated above, loss-of-function mutations in *ATRX* likely play central pathogenic roles in several distinct tumor variants, including multiple subtypes of incurable glioma. That *ATRX* itself encodes a chromatin regulatory protein suggests that epigenomic dysfunction underlies, at least in part, the oncogenic sequelae of its inactivation. To this point, we recently demonstrated that ATRX deficiency induces extensive chromatin remodeling and transcriptional shifts in putative glioma cells of origin, driving disease-relevant phenotypes that modulate both cellular motility and differentiation[44]. However, the full impact of ATRX deficiency on tumor initiation and evolution almost certainly includes other molecular mechanisms. The association of *ATRX* mutation and ALT[17,22], for instance, is now extensively described and provides a vehicle to telomerase-independent immortalization in affected cancer cells. Moreover, recent work has linked ALT to the well-characterized genomic instability induced by ATRX deficiency[16,45].

The pathogenic consequences of ATRX-dependent genomic instability in the context of cancer are unknown. Abundant prior work has demonstrated links between ATRX deficiency, DNA damage, CNA development, and aneuploidy[18–23]. Indeed,

p53-dependent apoptosis derived from genomic instability in the neuroepithelial progenitor compartment likely underlies the neuronal depopulation, microcephaly, and mental retardation associated with ATR-X syndrome[46]. Replication stress has been extensively implicated as a root cause of genomic instability in ATRX-deficient cells[19,21,47]. In addition to activating DNA damage pathway signaling, replication fork stalling and collapse can generate double-strand breaks and defective chromosome condensation during mitosis, both of which are known to drive CNAs and aneuploidy of the kind seen in *ATRX*-mutant glioma[48–50]. Recent work strongly supports the notion that the replication stress characterizing ATRX-deficient cells derives, at least in part, from G4 DNA secondary structure[16,27–29]. ATRX binds widely at GC-rich genomic sites susceptible to forming G4s[25], and restored ATRX expression in *ATRX*-mutant cell lines mitigates G4-associated phenotypes such as ALT[16]. Such data imply that ATRX may serve to protect the genome from unwanted G4 formation and the potentially deleterious consequences of ensuant genomic instability. Our findings support this conjecture by demonstrating, for the first time, that ATRX deficiency potently and reversibly induces G4 formation in isogenic experimental models ranging from transformed astrocytes

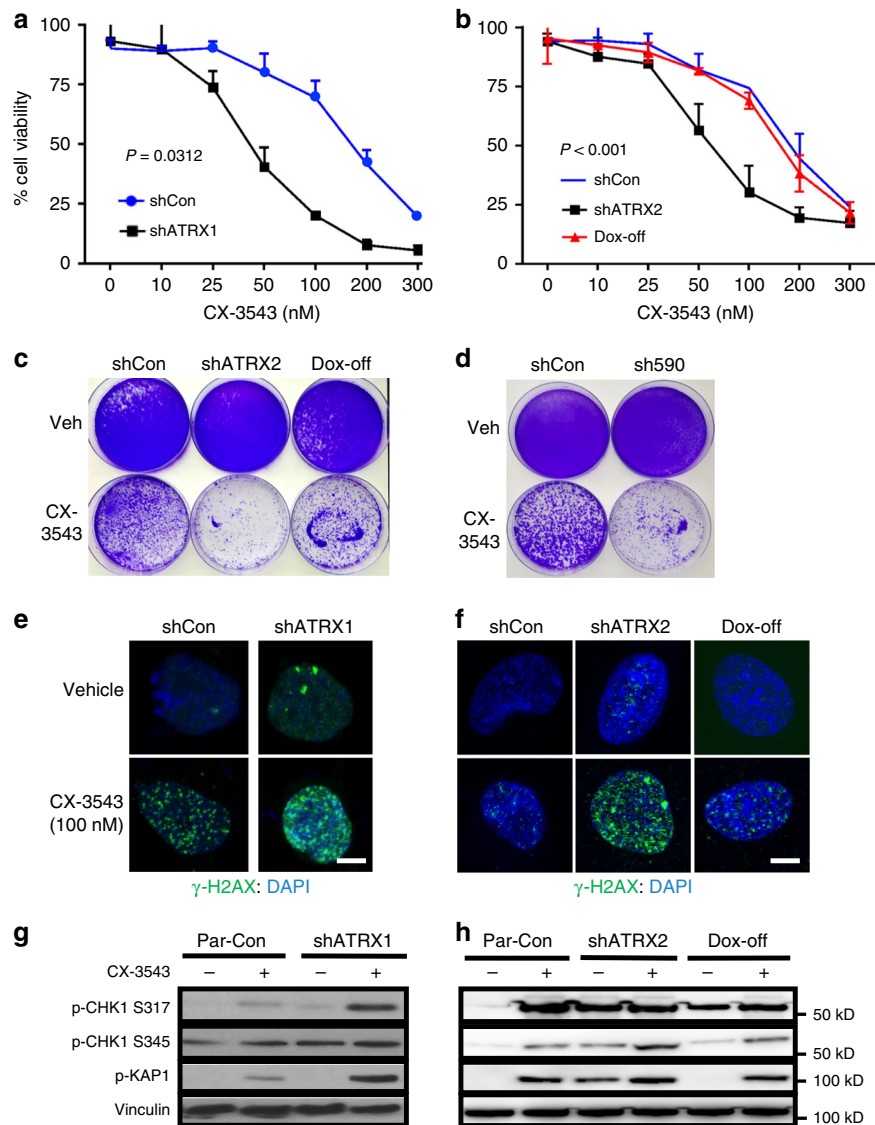

**Fig. 5** ATRX-deficient glioma models are selectively sensitive to G4 stabilization. **a**, **b** Cell viability (CellTiter-Glo) of constitutive (**a**) and inducible (**b**) shCon and shATRX NHAs (four replicates each) treated with CX-3543 from 0-300nM. **c** Clonogenic assay of inducible shATRX2 NHAs demonstrates enhanced and reversible sensitivity to CX-3543 (50 nM) with ATRX deficiency. **d** Clonogenic assay of TS 543 with (sh590) and without (shCon) ATRX knockdown demonstrates enhanced sensitivity to CX-3543 (50 nM) with ATRX deficiency. **e**, **f** γ-H2AX immunofluorescence in constitutive (**e**) and inducible (**f**) shATRX NHAs showing increased DNA damage with CX-3543 treatment (100 nM), particularly in the setting of ATRX knockdown. **g**, **h** Western blots showing increased phosphorylation of replication stress pathway constituents (CHK1 and KAP1) in constitutive (**g**) and inducible (**h**) shATRX NHAs following CX-3543 treatment (100 nM). Where applicable, error bars reflect SEM; $P$ values determined by two-way ANOVA; scale bars represent 10 μm

to patient-derived GSCs. As such, they confirm a novel functionality for a SWI/SNF epigenetic regulator already widely implicated in chromatin remodeling, structure, and organization.

That increased G4s were accompanied by replication stress signaling, DNA damage at spatially overlapping sites—as confirmed by both immunofluorescence and ChIP, and disease-relevant patterns of CNAs in our cell line models provides additional evidence that this pathobiological cascade features in ATRX-mutant neoplasia. Prior computational analysis across multiple tumor types established significant correlations between CNA breakpoints and genomic sites enriched in putative G4-forming sequences[51], firmly implicating G4s in the process of cancer-associated CNA acquisition. In our NHA models, ATRX knockdown led to a distinct CNA profile over time enriched in alterations over 1 Mb in size. While this pattern did not

completely mirror the known CNA signature of *ATRX*-mutant gliomas[2], it did recapitulate key elements involving larger, arm-level events. In particular, two CNAs (12p gain and 14q loss) were reminiscent of analogous alterations in human tumors associated with unfavorable prognosis. These data speak directly to the premise that CNA mobilization, driven at least in part by G4-mediated DNA damage, promotes malignant evolution in ATRX-deficient gliomas. As this tumor subtype characteristically progresses slowly over time[52], such mechanistic insights are consistent with established clinical features.

Due to its sheer prevalence in glioma, ATRX deficiency represents a molecular target of intriguing therapeutic potential. That being said, effective strategies to drug an inactivated epigenetic regulator are not immediately obvious, as they might be in the setting of more conventional, kinase-predominant,

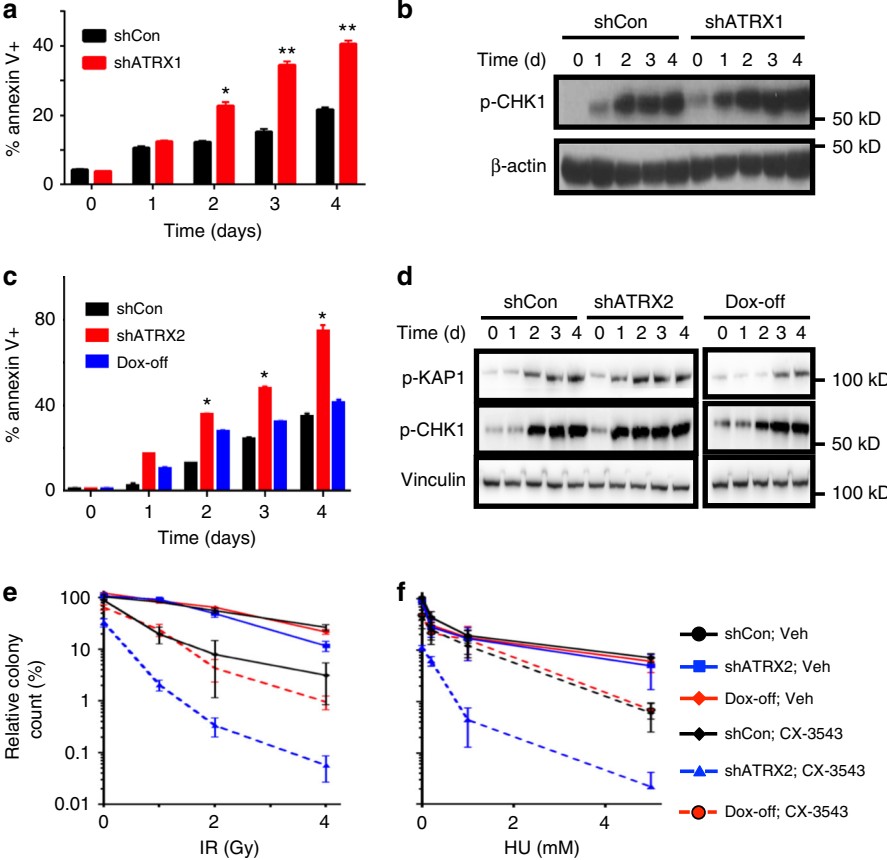

**Fig. 6** G4 stabilization selectively promotes apoptosis and synergizes with DNA-damaging therapies. **a, b** Time course study in shCon and shATRX1 NHAs (three replicates each) treated with 100 nM CX-3543 showing parallel kinetics of apoptosis (Annexin V positivity) (**a**) and p-CHK1/p-KAP1 levels by western blot (**b**). **c, d** Analogous time course study in inducible shATRX2 cells (three replicates each) documenting p-CHK1 an p-KAP1 levels (**d**) by western blot and Annexin V positivity (**c**) over time. **e, f** Soft agar colony counts for NHAs (3 replicates each) treated with either vehicle control (Veh) or 50 nM CX-3543 and either increasing doses of IR (**e**) or HU (**f**). Colony counts measured at 21 days scaled to that of shCon, Veh NHAs. Where applicable, error bars reflect SEM; $P$ values determined by two-way ANOVA (*$P < 0.05$, **$P < 0.01$)

oncogenic signaling networks. Given these challenges, leveraging specific vulnerabilities engendered by ATRX loss might offer alternative approaches. In particular, the longstanding association of ATRX deficiency with genomic instability, confirmed in this report, presents a tangible opportunity to explore synthetic lethality paradigms, akin to that of poly (ADP-ribose) polymerase (PARP) inhibitors in the treatment of *BRCA1*-inactivated breast cancer[53]. While the observed level of DNA damage in our ATRX-deficient cell line and tumor models was insufficient to induce apoptosis in isolation, due in part to coincident *TP53* inactivation, we hypothesized that its targeted enhancement would overwhelm compensatory mechanisms maintaining cell viability (Fig. 9). Moreover, our identification of G4s as the likely source of ATRX-deficient genomic instability provided a viable approach to therapeutic selectivity.

We found that G4 stabilization with multiple distinct agents selectively targeted ATRX-deficient glioma cell lines and tumors, both in vitro and in vivo. That our findings, initially obtained with CX-3543, were recapitulated with CX-5461, and PDS argues that on-target effects dependent on G4 binding were chiefly responsible for observed therapeutic impact. Moreover, cell death in these contexts was temporally associated with DNA damage and replication stress, further supporting impaired G4 resolution as a likely mechanism of action. These results recapitulate recent data showing enhanced sensitivity of ATRX-deficient embryonic stem cells to CX-5461[24]. We cannot completely exclude the

possibility that G4-stabilzation exerts some of its cytotoxic effects through the manipulation of ALT. As alluded to above, prior work has functionally linked increased G4s and DNA damage at telomeres with ALT induction in ATRX-deficient cells[16]. Nevertheless, ATRX knockdown was not associated with ALT in our NHA and TS 543 isogenics, consistent with multiple prior reports[19,20,22], and CX-3543 failed to alter the pattern of telomere FISH in *ATRX*-mutant JHH-273 GSC xenografts. Taken together, these findings strongly suggest that the cytotoxicity of G4 stabilization in the ATRX-deficient context is, at least in large part, mediated by DNA damage genome-wide, not limited to telomeric regions.

We also demonstrated that G4 stabilization dramatically enhanced the effects of IR and HU in ATRX-deficient NHAs, highlighting possibilities for effective synergistic combinations in the clinical setting. Since its introduction almost 40 years ago, IR has remained one of the most important nonsurgical therapeutic modalities employed in the treatment of malignant glioma, with demonstrated efficacy across disease subtypes[54–56]. Moreover, recent work has shown that ATRX-mutant gliomas in particular exhibit increased sensitivity to DNA-damaging combinations of IR and chemotherapy[57,58]. This vulnerability may derive in part from increased genomic instability at baseline. Defective non-homologous end joining (NHEJ), documented to arise with ATRX deficiency in preclinical models[20], may also play a role. Regardless of the precise molecular mechanisms at

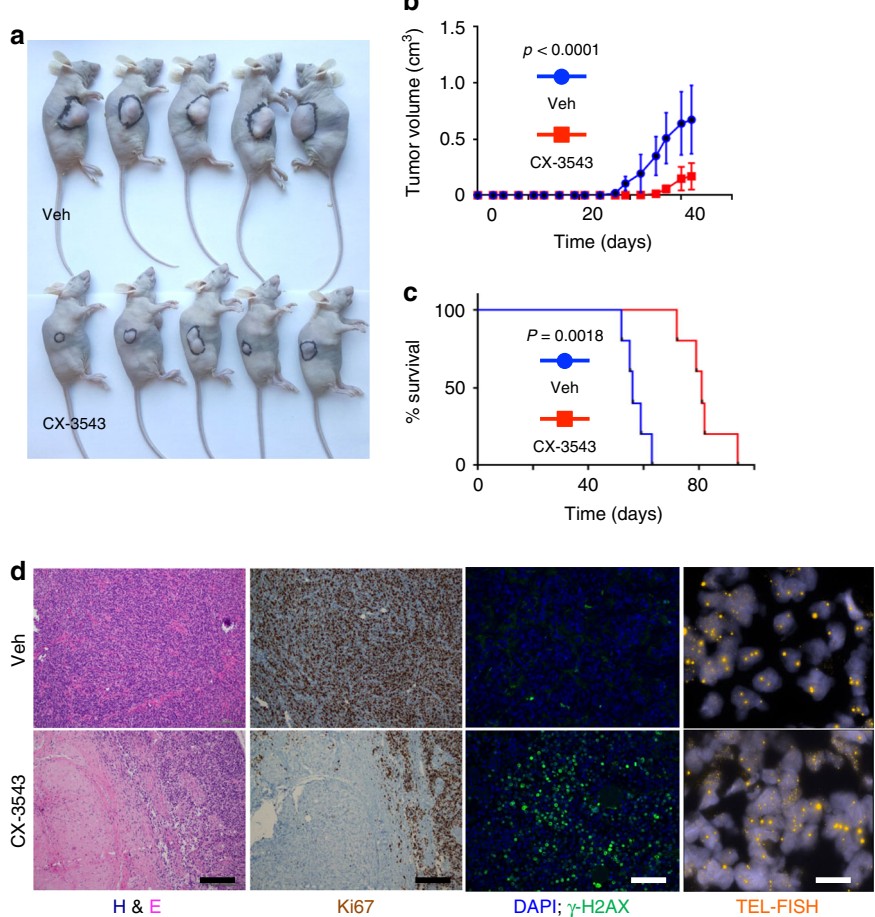

**Fig. 7** G4 stabilization markedly slows the growth of ATRX-mutant glioma xenografts. **a** representative image of mice bearing JHH-273 xenografts following treatment with either vehicle (veh) or 12.5 mg/kg CX-3543 for 42 days. **b** Average xenograft volume over time in 10 mice randomized (5 and 5) into two groups, receiving either 12.5 mg/kg CX-3543 or vehicle control (veh). **c** Kaplan–Meier analysis of survival in a separate cohort of 10 mice randomized (5 and 5) into two groups, receiving either 12.5 mg/kg CX-3543 or vehicle control (Veh). **d** Representative H&E-stained, immunostained (Ki67 and γ-H2AX), and TEL-FISH stained sections of xenografts from vehicle (Veh) and CX-3543-treated mice. CX-3543-treated tumors showed decreased cellularity, decreased proliferative activity, and increased DNA damage. *P* values were determined by two-way ANOVA; scale bars represent 200 μm in staining panels and 25 μm in Tel-FISH panels

work, therapeutically potentiating an already effective treatment strategy for glioma represents an underexplored approach with the potential for considerable clinical impact.

Precisely which G4 stabilizer represents the optimal agent for clinical translation remains unclear. Both CX-3543 and CX-5461 have advanced to clinical trials for pancreatic neuroendocrine tumor and BRCA1/2-deficient breast cancer, respectively[40,42]. However, CX-3543, by report, has limited bioavailability[41], and no formal blood–brain barrier penetration studies have been released for either. Nevertheless, our findings indicate that the targeted approach of G4 stabilization has considerable therapeutic potential in the treatment of ATRX-deficient glioma, along with other *ATRX*-mutant cancers. That the strategy is based on a tumor-specific vulnerability arising in association with an easily assessable biomarker should facilitate its clinical application, while also minimizing harmful side effects in treated patients. Moreover, alternative G4-stabilizing agents are currently available for use both as tool compounds and starting points for chemical derivatization[42,59,60].

In summary, we firmly implicate G4 secondary structure as a defining characteristic of *ATRX*-mutant glioma, one that drives disease-relevant genomic instability and presents opportunities

for tangible therapeutic advancement. As such, our work has important implications for both the molecular pathogenesis of ATRX-deficient neoplasia, as well as the development of more effective drugs specifically targeting a palette of deadly tumors.

## Methods

**Study design**. The objective of this study was to determine the impact of ATRX deficiency on G4 formation, DNA damage, and genomic instability in glioma, and assess the potential of chemical G4 stabilization as a therapeutic strategy in ATRX-deficient tumors. This was a controlled, laboratory-based, experimental study using cell line models in culture and in xenografts. ATRX was inactivated by genetic approaches and, in some cases, pharmaceutical agents and/or ionizing radiation were applied. Sample sizes were determined independently for each experiment without formal power calculation. No data was excluded from analysis. Unless otherwise specified, experiments employed three replicates per sample. End points varied by experiment and are described below, in figure legends, or in the Results section. Histopathological and immunohistochemical review of xenografts was conducted by a Neuropathologist (J.T.H.) in a nonblinded fashion. Quantification of G4 and/or γ-H2AX immunostaining in NHAs was blinded.

**Antibodies**. All commercially available antibodies used in this study, along with their source and application(s), are listed in Supplementary Table 1.

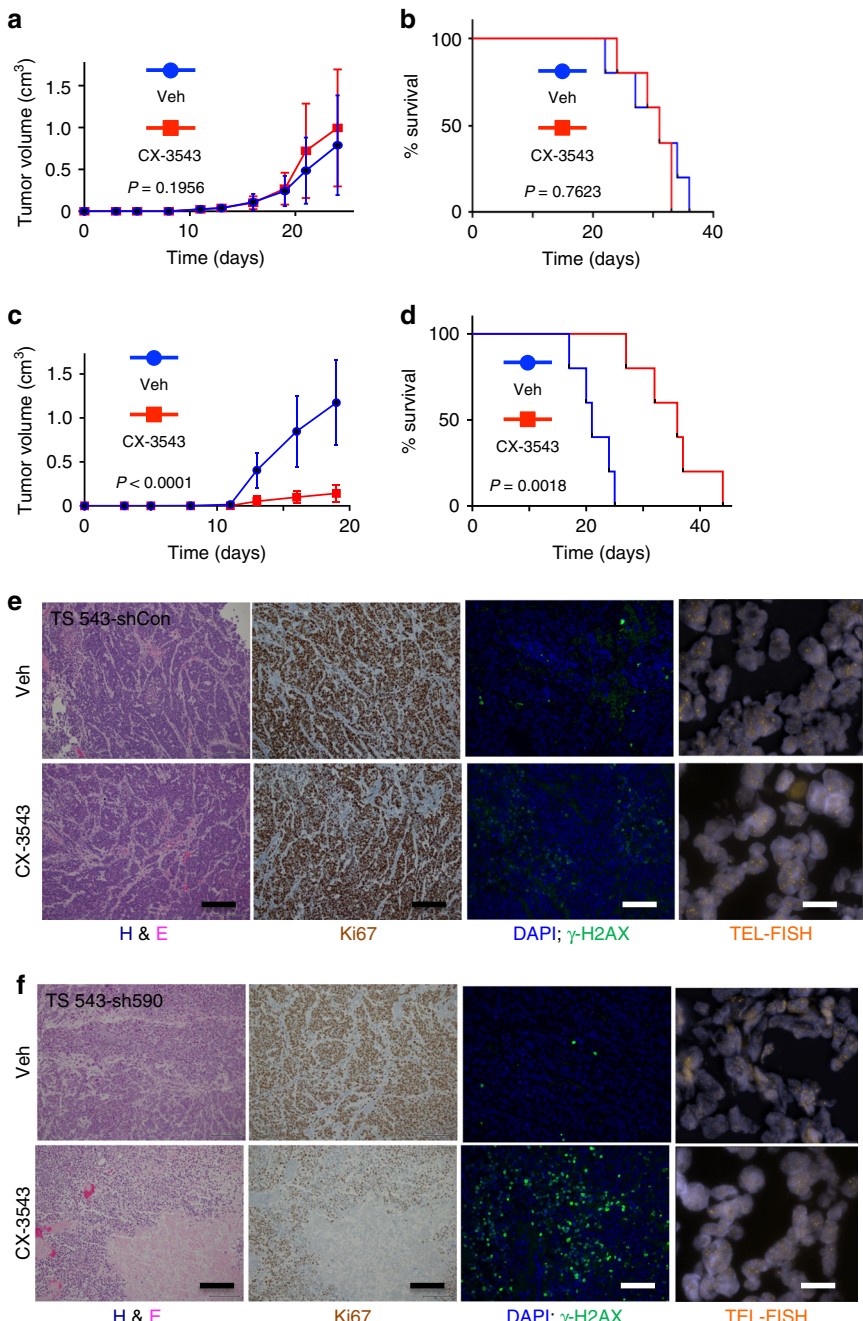

**Fig. 8** Efficacy of G4 stabilization in vivo is dependent on ATRX deficiency. **a** TS 543 (ATRX intact) xenografts exhibited similar rates of growth when treated with either vehicle control (Veh, $n = 3$) or 12.5 mg/kg CX-3543 ($n = 3$) as reflected by tumor volume over time. **b** Kaplan–Meier analysis of survival In a separate cohort of 10 mice randomized (5 and 5) into two groups, receiving either 12.5 mg/kg CX-3543 or vehicle control (Veh). **c** ATRX knockdown in TS 543 cells (sh590) restored the sensitivity of xenografts to CX-3543 treatment as shown by tumor volume over time in 10 mice randomized (5 and 5) into two groups, receiving either 12.5 mg/kg CX-3543 or vehicle control (Veh). **d** Kaplan–Meier analysis of survival In a separate cohort of 10 mice randomized (5 and 5) into two groups, receiving either 12.5 mg/kg CX-3543 or vehicle control (Veh). **e, f** Representative H&E-stained, immunostained (Ki67 and γ-H2AX), and TEL-FISH stained sections of xenografts from vehicle (Veh) and CX-3543-treated mice harboring either TS 543-shCon (**e**) or TS 543-sh590 (**f**) xenografts. CX-3543-associated histopathological effects were limited to TS 543-sh590 xenografts. Evidence of ALT was not seen in either GSC line. $P$ values were determined by two-way ANOVA for tumor growth and Log-rank (Mantel–Cox) test for survival curve comparison; scale bars represent 200 μm in staining panels and 25 μm in Tel-FISH panels

**Cell culture and generation of ATRX-deficient cell lines**. All cell lines used in this study were tested for mycoplasma contamination every three months at the MSKCC Antibody and Bioresource Core. Parental immortalized normal human astrocytes were a gift from R.O. Peiper (UCSF)[61]. TS 543, TS 603, and 08-0537 are patient-derived GSCs[62–64] maintained in NeuroCult™ NS-A Proliferation media (#05751, Stemcell). 08-0537 was generously provided by Hai Yan (Duke).

ATRX knockdown was achieved by introducing either a modified FUGW vector (a gift from David Baltimore (Addgene plasmid # 14883)) carrying an shRNA expression cassette against *ATRX* (shATRX1) (see Supplementary Table 2 for shRNA sequences), a TRIPZ TET-inducible vector (Dharmacon) containing a distinct shRNA against ATRX (shATRX2), or a third shRNA against ATRX (sh590) from the TRC shRNA library (Sigma). shATRX1- and shATRX2-positive

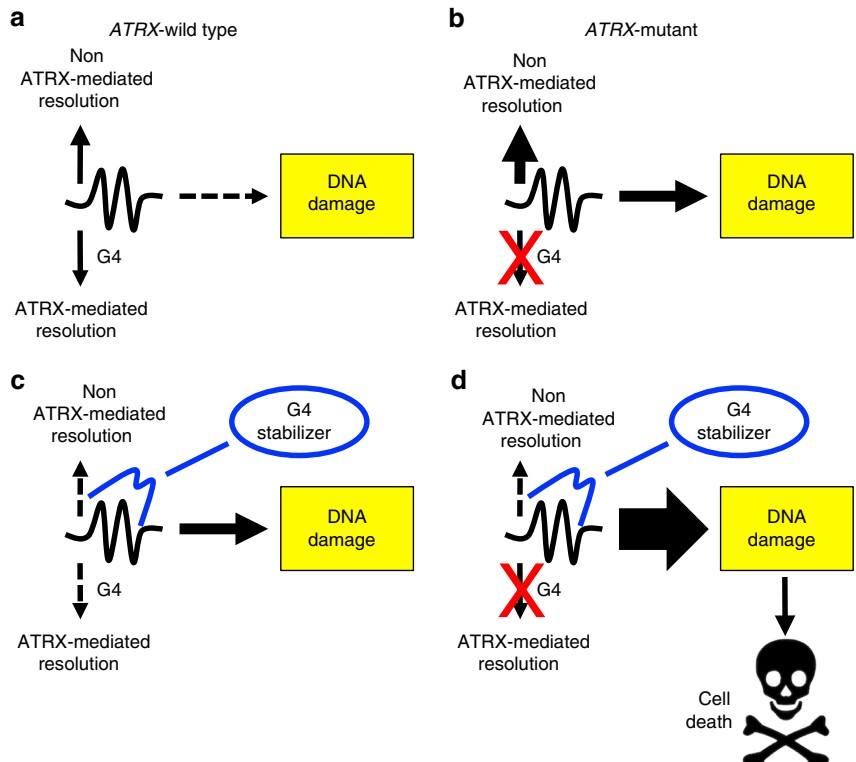

**Fig. 9** Selectively targeting G4s in the ATRX-deficient context. G4s are normally resolved by both ATRX-dependent and ATRX-independent mechanisms to mitigate DNA damage in cells (**a**). In the setting of ATRX deficiency, DNA damage increases but its cytotoxic effects are dampened by ATRX-independent G4 resolution, maintaining cellular viability (**b**). However, concurrent G4 stabilization impairs these salvage pathways (**c**, **d**) further enhancing DNA damage and inducing cell death in the ATRX-deficient context

cells were FACS-sorted every two passages by fluorescent marker (RFP) for the top 5% of total population to ensure high shRNA expression. Sh590-positive TS 543 cells were subjected to puromycin-based selection.

**Proliferation, cell cycle, and apoptosis analyses.** Flow-cytometry analyses of proliferation and cell cycle were performed using the BD Pharmingen BrdU Flow Kit (# 559619). Apoptosis assays were performed using the Dead Cell Apoptosis Kit (# V13241, Thermo Fisher) with Propidium Iodide (PI) substituted by DAPI to avoid RFP interference.

**In situ visualization of G-quadruplexes, γ-H2AX, 53BP1, and BLM.** The 1H6 antibody was a gift from Dr. Peter M. Lansdorp[65]. The BG4 antibody was purchased from Millipore (MABE917). For immunostaining, cells were grown in chamber slides (Nunc Lab-Tek II, cat no. 154526, Thermo Fisher) and synchronized to G0 phase by 24-h serum starvation. The cells were digested with 10 mg/ml proteinase K for 1h at 37 °C, followed by fixation (4% paraformaldehyde in PBS for 10 min) and permeabilization (0.5% Tween-20, 0.2% Triton X-100 in PBS, 10 min). To eliminate RNA-structures, cells were treated with 20 ug/500 ul RNase A (Invitrogen). To confirm specificity towards DNA-G4, cells were incubated in 40 mM Tris Cl (pH 8), 5 mM CaCl2, 2 mM MgCl2, 100 ug/ml BSA alone or including 0.06 U/ul of DNase I (Promega) and 80 gel units/ul of micrococcal nuclease (#M0247S, New England Biolabs) at 37 °C for 2 h. For staining, cells were blocked with goat serum (Sigma) for 4 h at room temperature, then incubated with 1H6 (0.5 μg/ml) or BG4 (1:100) at 4 °C overnight. Slides were then washed five times with PBST, incubated with Alexa Fluor 488 or 568 goat anti-mouse IgG (Invitrogen) at room temperature for 2 h, washed five times with PBST and mounted with coverslips using ProLong Gold antifade reagent and DAPI counterstain (Invitrogen). For γ-H2AX monostaining or 53BP1/1H6 or γ-H2AX/BG4 double staining, cells were treated with or without CX-3543 (100 nM), CX-5461 (50 nM), or PDS (2 μM) for 3 days prior to synchronization to G0 and incubation with the appropriate primary antibody combinations at 4 °C overnight (γ-H2AX antibody (1:500): # 05-636, Millipore; 53BP1 antibody (1:1000): cat# NB100-304, Novus Biologicals). Secondary antibodies included goat anti-mouse or goat anti-rabbit Alexa Fluor 488 or 568 (1:2000), as appropriate, and were applied as described above. For BLM/γ-H2AX double staining, fixation/permeabilization was performed in ice-cold 100% methanol, and staining conducted sequentially with γ-H2AX antibody (1:500, overnight) and Alexa Fluor 568 goat anti-mouse

antibody (1:2000, 2 h), followed by BLM antibody (Bethyl, A300-110A, 1:50) (overnight) and Alexa Fluor 488 goat anti-rabbit antibody (1:2000, 2 h).

**G4 pulldowns.** Plasmid expressing hf2 was a kind gift of Dr. Shankar Balasubramanian[66,67]. Hf2 antibodies were expressed in Bl21 competent cells subjected to 1 mM IPTG induction. The culture supernatant was then bound to Protein A Sepharose (#P9424, Sigma) by chromatography, followed by washing with 50 mM potassium phosphate, 100 mM potassium chloride buffer (pH 7.4) three times. Antibody was then eluted with 0.1 M Tricine buffer (pH 3.0) into 0.1 M potassium phosphate buffer (pH 8.0). For G4 pulldowns, 2 μg of hf2 and 50 μl of Protein A Dynabeads (#10001D, Thermo Fisher) were mixed and incubated overnight rotating at 4 °C. Beads were washed with PBS five times. Ten μg of genomic DNA from NHAs was sonicated and incubated with beads in 0.5% BSA overnight rotating at 4 °C, followed by six washes with 10 mM Tris pH 7.4, 100 mM KCl, 0.1% Tween-20 and one wash with 10 mM Tris pH 7.4, 100 mM KCl. Bound DNA was eluted in 50 μL of 1% SDS, 0.1 M NaHCO3 at 30 °C for 1 h then purified by QIAquick PCR Purification Kit (Qiagen) to a final volume of 20 μL. The recovered DNA was used to determine enrichment of telomeric sequence (Tel1, 2 and X) and the promoter regions of *MYC* and *ZNF618*, using the *ESR1* promoter as a negative control (See Supplementary Table 3 for primer sequences).

**Chromatin immunoprecipitation (ChIP).** NHAs (shCon and shATRX1, $3 \times 10^7$ cells) were cross-linked with formaldehyde (0.75% v/v, RT, 15 min), quenched with 125 mM glycine (RT, 5 min), harvested, and sonicated with a Bioruptor (Diagenode) in 50 mM HEPES-KOH, pH 7.5, 140 mM NaCl, 1 mM EDTA pH 8, with 0.1% SDS and proteinase inhibitors, to obtain DNA fragments of 200–300 bp[68]. Fragmented DNA was then subjected to immunoprecipitation with γ-H2AX (2 μg) and BLM (5 μg) antibodies (see above) and IgG controls at 4 °C overnight with constant agitation. DNA-antibody complexes were then incubated with 20 μl Magna Protein G Magnetic Beads (#16-662, Millipore) overnight at 4 °C with constant agitation. Recovered DNA fragments were measured for enrichment at *MYC*, *ZNF618*, and *ESR1* promoters as described above. Additionally, shATRX1 NHAs were treated with PDS (2 μM) for 3 days before fixation and ChIP, and enrichment at *MYC*, *ZNF*, and *ESR1* loci was compared to that seen in vehicle treated shATRX1 NHAs.

**TEL-FISH and metaphase cytogenetic analysis**. For cell lines, resuspended cells were incubated with Colcemid (0.1 μg/ml) at 37 °C for 45 min, resuspended in 0.075 M KCl and incubated at 37 °C for 10 min, followed by fixation in methanol: acetic acid (3:1) solution. TEL-FISH was performed according to standard procedures using a CY3-conjugated, telomere-specific nucleic acid probe: 5'-TTAGGGT TAGGGTTAGGG-3' (Applied Biosystems). For xenograft tissues, tumors were removed and subjected to OCT embedding followed by 5 μm sectioning. Frozen sections were fixed with 4% paraformaldehyde for 10 min. After denaturing at 85 °C for 5 min in 10 mM Tris-HCL pH 7.2, 70% formamide, 0.5% blocking solution reagent (Roche), hybridization was performed as described above.

**Cell viability and clonogenic assay**. For standard viability assays, cells (500/well) were incubated with a serial concentration of CX-3543 (10–300 nM), CX-5461 (2.5–500 nM), or PDS (0.1–20 μM) for 7 days in 96-well plates. Cell viability was then assessed with the CellTiter-Glo Luminescent Assay (Promega) according to manufacturer-recommended procedures. To determine clonogenic ability, NHA or TS 543 cells were seeded at 5000 cells/10-cm dish and incubated with vehicle or 50 nM CX-3543 for 14 days. Cells were fixed with 4% paraformaldehyde and stained with 0.005% crystal violet in PBS, followed by three washes in PBS and two washes in ddH₂O. For soft agar colony formation assays, 50,000 cells were seeded in 6-well plates containing 1% bottom layer and 0.5% top layer soft agar. Cells were then cultured in growth media with or without 50 nM CX-3543. Radiation dosing of 0, 1, 2, or 4 Gy was immediately applied after plating. The 1.5 ml growth media covering the agar cultures was replenished every week. At day 21, colonies were fixed with 4% paraformaldehyde for 30 min and stained with 0.005% crystal violet in PBS overnight. Stained colonies were then washed extensively in PBS and water, and quantified on a Gelcount colony counter (Oxford Optronix).

**SNP arrays**. Genomic DNA was isolated from ATRX-deficient NHAs at passages 5 and 15. As controls, genomic DNA from ATRX-intact parental NHAs was derived at the start point (P0), P5, and P15. Extracted DNA was subjected to Affymetrix Genome-Wide Human SNP 6.0 array analysis (cat# 901182, Thermo Fisher) according to the manufacturer's protocol. Preliminary copy number derivation was facilitated by circular binary segmentation[69,70] to generate CNV segment files with the following information: chromosome, start position, end position, probe number, and segment mean value. For analysis, we focused variations with absolute segment mean value >0.5 for LGG samples and >0.1 for NHA lines. All variations associated with ChrX and ChrY were excluded. CNV length was calculated by using the end position minus the start position. Data were visualized using IGV and GISTIC2.0.

**Xenograft experiments**. All animal protocols and procedures were performed in the xenograft suite at Memorial Sloan-Kettering Cancer Center (Animal protocol # 07-09-015) in accordance with the ethical and experimental regulations of the Institutional Animal Care and Use Committee (IACUC). JHH-273 samples were kind gifts from Dr. Gregory Riggins at the Johns Hopkins University. Tumor samples were mechanically dissociated and small pieces (0.2 mm³) were embedded into the flanks of nude mice (Taconic Farms). In parallel, ATRX-intact and ATRX-deficient TS 543 cells at exponential growth phase were dissociated with Accutase (#07920, Stemcell), resuspended in Neurocult media, mixed with Matrigel (#356234, Corning) (1:1) and injected into nude mice flanks in a 50 μl mixture containing 5 × 10⁶ cells. Mice were randomized to vehicle or CX-3543 (12.5 mg/kg) treatment groups. Drug delivery occurred via intravenous injection once per day, on a 5 day/week schedule until health-related defined end points. Tumor volumes were measured by calipers and calculated using the formula (l × w²)/2, where w is width and l is length in mm. For survival experiments, mice were treated until they reached health-related end points (2000 mm³ tumor volume). For growth curve comparisons, all mice in a study cohort were sacrificed when the first mouse reached the 2000 mm³ tumor size threshold. An independent cohort was used for Kaplan–Meier analysis. Xenografted tissues were removed, weighted, and split into two parts. One part was snap frozen for TEL-FISH, while the other half was subjected to FFPE processing. five-micrometer FFPE sections were deparaffinized and subjected to antigen retrieval. Sections were blocked for non-specific binding with goat serum for 2 h, followed by staining with Ki67 (5μg/ml, ab15580, Abcam) or γ-H2AX (1:1000, # 05-636, Millipore) antibodies at 4 °C overnight. Sections were washed and incubated with secondary antibody. Ki67 staining were counterstained with Hematoxylin, and γ-H2AX staining were counterstained with DAPI.

**Statistics**. Unless otherwise stated, all results, representing at least three independent experiments, were plotted as mean ± SEM. In general, data were statistically analyzed using unpaired, two-tailed t-tests. Log-rank (Mantel–Cox) test were used to determine the significance of differences in Kaplan–Meier analysis of LGG patients and of hind flank xenograft experiments. Two-way ANOVA was used to compare the growth curves of xenografts and the colony formation assays. P values are represented using * for P < 0.05, ** for P < 0.01, *** for P < 0.001, and **** for P < 0.0001.

**Reporting summary**. Further information on experimental design is available in the Nature Research Reporting Summary linked to this article.

**Code availability**. No customized code was used for data processing

## Data availability

All data (raw and processed) and materials related to this manuscript will be made available upon request, utilizing material transfer agreements when appropriate. Raw SNP array data and copy number variation profiles have been deposited in Gene Expression Omnibus (GSE125296), [https://www.ncbi.nlm.nih.gov/geo/query/acc.cgi?acc=GSE125296]. Raw western blot data are presented in Supplementary Fig. 12.

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

## Acknowledgements

We would like to thank Drs. Peter Lansdorp and Shankar Balasubarmanian for providing the 1H6 and hf2 antibodies, respectively. We would also like to acknowledge Dr. Cameron Brennan for providing the TS 543 GSC line. Finally, we would like to acknowledge the Genomics Core Facility at the Albert Einstein College of Medicine for their assistance in performing SNP arrays, and the Molecular Cytogenetics Core at MSKCC for Tel-FISH. J.T.H. is supported by a Research Scholars Grant, RSG-16-179-01-DMC, from the American Cancer Society. Support for this work also came from the Sontag Foundation (J.T.H.), the Sidney Kimmel Foundation (J.T.H.), and Cycle for Survival (J.T.H.). We acknowledge support from two NIH/NCI Cancer Center Support Grants (CCSGs) for MDACC (P30 CA016672) and MSKCC (P30 CA08748).

## Author contributions

Conceptualization and design were done by Y.W., T.A.C. and J.T.H. Development and methodology were done by Y.W., T.A.C. and J.T.H. Acquisition of data was done by Y.W., A.T.W., W.H.W., R.S. and C.D. Analysis and interpretation of data were done by Y.W., J.Y., K.K., T.A.C. and J.T.H. Writing, review, and/or revision of the manuscript were done by Y.W., K.K., T.A.C. and J.T.H. Administrative, technical, or material support were done by G.J.R., K.K., E.P.S., T.A.C. and J.T.H. Study supervision was done by T.A.C. and J.T.H.

## Additional information

**Competing interests:** T.A.C. is a co-founder of Gritstone Oncology and holds equity. He holds equity in An2H. He acknowledges grant funding from Bristol-Myers Squibb, AstraZeneca, Illumina, Pfizer, An2H, and Eisai, and he has served as a paid advisor for Bristol-Myers Squibb, Illumina, Eisai, and An2H. M.S.K. has licensed the use of TMB for the identification of patients that benefit from immune checkpoint therapy to PGDx and T.A.C. receives royalties as part of this licensing agreement. The remaining authors declare no competing interests.

