## [Peer Review File · Nature Communications]

Editorial Note: Parts of this Peer Review File have been redacted as indicated, in the interest of confidentiality relating to personal correspondence.

Reviewers' comments:

Reviewer #1, Expertise: G4 quadruplexes, ATRX (Remarks to the Author):

G-quadruplex DNA drives genomic instability and represents a targetable molecular abnormality in ATRX-deficient malignant glioma

This is an interesting report which sets out to establish if genome instability in ATRX deficient neuronally derived cells is related to the presence of G quadruplex (G4) structures, which ATRX may be involved in resolving, and that this provides a process that may be exploited therapeutically through the use of G4 stabilisers. If proved correct this will provide one of the missing links in understanding how ATRX acts as a tumor suppressor as well as a rational therapy for glioma.

1. A major problem, however, is that the data presented does not prove this link beyond question. One problem is the lack of reagents that identify G4 structures in vivo. Although a number have been published such as antibodies IH6, hf2 and BG4 there has been a paucity of subsequent reports confirming their bone fides. Given this, the data involving such reagents has to be cast iron to be convincing.

IH6 was raised against an (TTTTGGGG)₂ oligo and the lab associated with the original report have subsequently shown that it does not bind to G4 structures without thymidines but cross reacts with thymidine rich ssDNA and probably is not recognising the G4 itself. (Kazemier HG, Paeschke K, Lansdorp PM. *Nucleic Acids Res.* 2017 45:5913-5919). Given this the data using this antibody needs to be repeated with alternative reagents. In some cases they have used hf2 but should, in addition, use BG4 to confirm their findings.

2. Only one piece of data is presented to show that the DNA damage observed in ATRX-deficient cells is associated with the presence of G4 (Figure 2c) and this is possibly using IH6 (not defined in the figure) and is by IF. There needs to be a higher level of proof that in these ATRX-deficient cells the sites of DNA damage (and presumably fragile sites seen at metaphase) are at sites of G4. I'm not confident that the G4 reagents available are good enough to show this by ChIP but one would expect to see G4 processing proteins such as BLM, WRN and PIF1 at these sites and this could be checked at the DNA sequence level by ChIP.

3. A problem with many G4-stabilising agents is that they cause DNA damage irrespective of their G4 binding activity. Furthermore there is evidence from a number of different reports that ATRX plays a role in DNA double strand break (DSB) repair. The exacerbation of DNA damage seen in the ATRX knockdown after treatment with the G4 stabilise CX-3543 does not necessarily prove that ATRX and this agent are acting at sites of G4 it might be secondary to the generation of DSBs. They should repeat the experiments using a new generation of G4 ligands with have lower toxicity as reported by Guilbaud 2017 *Nat Chem* 9, 1110. This at least would strengthen their conclusions if the findings are confirmed.

Minor points:

1. A number of reports have described the development of osteosarcoma in individuals with germline mutations in ATRX (Masliah-Plachon et al 2018 *EurJHumGenet*; Ji et al 2017 *AmJMedGenet*; Smolle et al 2017 *Ped Blood Cancer*). The statement about these individuals not being predisposed to cancer needs to acknowledge these reports.

2. They should reference the recent report by Udugama 2018 in *PNAS* on rDNA instability in ATRX knockout.

3. In Fig 1e it is rather counterintuitive that ATRX does not co-localise with G4 if involved in

resolving them.

4. They should state how many metaphases were scored in the chromosome break figure 2e. The chromosome fragility seen in this figure has not previously been reported - do the authors know if it is cell type specific?

5. In Figure 4 sh590 should be explained in the legend rather than just in M&Ms.

6. In Supp Fig1 is "untreated" in s1b meant to refer to "treated with CX-3543"?

7 In Suppl fig 2 it is not clear what the panels in s2a are demonstrating - I presume it is the lack of regions of intense telomeric signal seen in ALT. Most readers will not have a clue about this. They should also include the shCon treated cells for comparison. There is no mention in the legend of the arrows in the panels in s2b.

Reviewer #2, Expertise: Gliomas, ATRX (Remarks to the Author):

This manuscript by Wang et al, report that G-quadruplex DNA drives genomic instability and represents a targetable molecular abnormality in ATRX-deficient malignant glioma. Overall, the data reported are interesting, the experiments are well designed and for the most part, the conclusions drawn by the authors are derived from the experimental data presented. Please find enclosed below several comments which need to be addressed by the authors:

1- It would make the data more clinically relevant if the authors assessed the formation of G4s in human glioma cells which are ATRX deficient. The authors have one such cell culture available as they have used it for the in vivo studies.

2- Also, the authors should comment if the effects they see in the NHA cells ATRX deficient reported in their manuscript would hold for all ATRX deficient glioma cell types independently of other co-expressed genetic lesions; or if the presence of other genetic alterations would have an impact on the phenotype they describe.

3- The rationale for the comparisons shown in Figure 3 need to be better justified/explained by the authors in the context of molecular glioma subtypes.

4- Data reported in Figure 5 should be repeated using the patient derived human glioma cells which are ATRX deficient.

5- The authors could include a diagram related to the proposed mechanism of action of the therapeutic intervention in ATRX mutant gliomas.

6- The authors should also revise the discussion in relation to the impact of IDH1 mutations on DNA repair (refs 53-55), as none of the papers cited, assess the impact of IDH1 mutation in the context of ATRX mutations, which is the main topic addressed in this manuscript. One of the cited papers is related to AML. the other paper used colon cancer cells and Hela transfected with mutant IDH1 and the third paper uses U87 and U251 cells transfected with mutant IDH1, none of these are relevant to the current manuscript nor to any relevant patient derived glioma cell culture expressing mIDH1 and harboring ATRX deficiency.

We would like to express our sincere gratitude to both reviewers for their insightful comments. In addressing their concerns, we feel that our revised manuscript is significantly strengthened. Please find a point-by-point response to reviewer comments below (reviewer comments italicized, responses in bold).

Reviewer #1:

This is an interesting report which sets out to establish if genome instability in ATRX deficient neuronally derived cells is related to the presence of G quadruplex (G4) structures, which ATRX may be involved in resolving, and that this provides a process that may be exploited therapeutically through the use of G4 stabilisers. If proved correct this will provide one of the missing links in understanding how ATRX acts as a tumor suppressor as well as a rational therapy for glioma.

1. A major problem, however, is that the data presented does not prove this link beyond question. One problem is the lack of reagents that identify G4 structures in vivo. Although a number have been published such as antibodies IH6, hf2 and BG4 there has been a paucity of subsequent reports confirming their bone fides. Given this, the data involving such reagents has to be cast iron to be convincing.

We greatly appreciate these sentiments, and have significantly strengthened the manuscript with additional studies employing alternative reagents, as described in more detail below.

IH6 was raised against an (TTTTGGGG)₂ oligo and the lab associated with the original report have subsequently shown that it does not bind to G4 structures without thymidines but cross reacts with thymidine rich ssDNA and probably is not recognising the G4 itself. (Kazemier HG, Paeschke K, Lansdorp PM. Nucleic Acids Res. 2017 45:5913-5919). Given this the data using this antibody needs to be repeated with alternative reagents. In some cases they have used hf2 but should, in addition, use BG4 to confirm their findings.

We appreciate this suggestion, and have now validated our major findings regarding increased G4s in ATRX-deficient NHAs and GSCs, and their colocalization with DNA damage sites, using BG4. These data are now presented in FIG. 1, Supplementary FIG. 1, Supplementary FIG. 3, and Supplementary FIG. 8, and described in the Results section, pp. 5-6, and 9.

2. Only one piece of data is presented to show that the DNA damage observed in ATRX-deficient cells is associated with the presence of G4 (Figure 2c) and this is possibly using IH6 (not defined in the figure) and is by IF. There needs to be a higher level of proof that in these ATRX-deficient cells the sites of DNA damage (and presumably fragile sites seen at metaphase) are at sites of G4. I'm not confident that the G4 reagents available are good enough to show this by ChIP but one would expect to see G4 processing proteins such as BLM, WRN and PIF1 at these sites and this could be checked at the DNA sequence level by ChIP.

We thank the reviewer for this insightful suggestion. In response, we have now performed ChIP/qPCR for both BLM and γ -H2AX in our NHA isogenics and documented increased enrichment for both at putative G4 sites in the promoter regions of MYC and ZNF618 arising with ATRX deficiency. Similarly, we find that

G4 stabilization in ATRX-deficient NHAs promotes further CHIP enrichment for BLM and γ -H2AX at these same sites. Enrichment is either modest or not significant at a control genomic site (*ESR1*). These findings provide further support for our conclusion that ATRX deficiency promotes DNA damage at sites of G4 formation. They are presented in FIG. 2 and Supplementary FIG. 9, are described in the Results section, pp. 7 and 9, and are briefly mentioned in the Discussion section, p. 13.

3. A problem with many G4-stabilising agents is that they cause DNA damage irrespective of their G4 binding activity. Furthermore there is evidence from a number of different reports that ATRX plays a role in DNA double strand break (DSB) repair. The exacerbation of DNA damage seen in the ATRX knockdown after treatment with the G4 stabilise CX-3543 does not necessarily prove that ATRX and this agent are acting at sites of G4 it might be secondary to the generation of DSBs. They should repeat the experiments using a new generation of G4 ligands with have lower toxicity as reported by Guilbaud 2017 Nat Chem 9, 1110. This at least would strengthen their conclusions if the findings are confirmed.

We appreciate this comment and have sought to validate our findings with other G4-stabilizing agents. The compound to which this reviewer refers (PDC12) is not readily available from commercial sources.

[REDACTED]

we procured PDS and employed it in analogous studies to those performed previously with CX-3543. We also recapitulated our experiments using another G4-stabilizing agent currently in clinical trials (CX-5461). We found that both agents increased DNA damage and cytotoxicity in ATRX-deficient NHAs relative to ATRX intact counterparts. Moreover, sites of enhanced DNA damage induced by PDS and CX-5461 spatially overlapped with G4s by both immunofluorescence and BLM/ γ -H2AX CHIP (IF studies with both agents, CHIP studies with PDS). These findings considerably strengthen our conclusion that G4-stabilization promotes DNA damage and cell death by way of on-target effects. These new data are presented in Supplementary FIG. 6-9 and are described in the Results section, p. 9, and the Discussion section, p. 14.

Minor points:

1. A number of reports have described the development of osteosarcoma in individuals with germline mutations in ATRX (Maslah-Plachon et al 2018 EurJHumGenet; Ji et al 2017 AmJMedGenet; Smolle et al 2017 Ped Blood Cancer). The statement about these individuals not being predisposed to cancer needs to acknowledge these reports.

We have stated these findings and cited the references listed above in the Introduction section, p. 3.

2. They should reference the recent report by Udugama 2018 in PNAS on rDNA instability in ATRX knockout.

We now describe and refer to this paper in both the Introduction section, p.3, and the Discussion section, p14.

3. In Fig 1e it is rather counterintuitive that ATRX does not co-localise with G4 if involved in resolving them.

As stated in the Results section, p. 6, we interpret this finding to mean that ATRX facilitates the resolution of G4 sequences to which it binds. This would render them non-reactive to antibodies specifically targeting the G4 structure.

4. They should state how many metaphases were scored in the chromosome break figure 2e. The chromosome fragility seen in this figure has not previously been reported - do the authors know if it is cell type specific?

We now report the number of cellular metaphase spreads (10) quantified for each analysis in the legend for FIG. 3 (new figure number due to revisions). This effect does not appear to be cell type specific. We now include data for both isogenic *Atrx*-intact and *Atrx*-KO murine neuroepithelial progenitor cells (mNPCs) and patient derived GSCs from *ATR*X-intact and *ATR*X-mutant tumors. In GSCs, increased chromosome breaks are associated with *ATR*X deficiency. A similar trend is also seen in mNPCs, although this particular finding does not reach statistical significance. These data are now presented in FIG. 3 and Supplementary FIG. 4, and described in the Results section, p. 7.

5. In Figure 4 sh590 should be explained in the legend rather than just in M&Ms.

This explanation is included in what is now the legend for FIG. 5d as follows: "Clonogenic assay of TS 543 with (sh590) and without (shCon) *ATR*X knockdown demonstrates enhanced sensitivity to CX-3543 (50 nM) with *ATR*X deficiency".

6. In Supp Fig1 is "untreated" in s1b meant to refer to "treated with CX-3543"?

Yes. We have revised the figure to make this clearer.

7. In Suppl fig 2 it is not clear what the panels in s2a are demonstrating - I presume it is the lack of regions of intense telomeric signal seen in ALT. Most readers will not have a clue about this. They should also include the shCon treated cells for comparison. There is no mention in the legend of the arrows in the panels in s2b.

We now include panels showing shCon NHA counterparts in this figure (Supplementary FIG. 4a) and describe the lack of ultrabright ALT foci in the accompanying legend. We also reference the white arrows in the legend for Supplementary FIG. 4.

Reviewer #2:

*This manuscript by Wang et al, report that G-quadruplex DNA drives genomic instability and represents a targetable molecular abnormality in *ATR*X-deficient malignant glioma. Overall, the data reported are interesting, the experiments are well designed and for the most part, the conclusions drawn by the authors are derived from the experimental data presented. Please find enclosed below several comments which need to be addressed by the authors:*

1- *It would make the data more clinically relevant if the authors assessed the formation*

of G4s in human glioma cells which are ATRX deficient. The authors have one such cell culture available as they have used it for the in vivo studies.

We appreciate this comment and have now included studies with isogenic IDH-wild type, ATRX-wild type GSCs (TS 543) with and without exogenous ATRX-knockdown (sh590), as well as IDH-mutant GSCs that are either ATRX-wild type (TS 603) or ATRX-mutant (08-0537). Using these lines, we have recapitulated our previous findings in isogenic ATRX-intact and ATRX-deficient NHAs showing increased G4s arising with ATRX deficiency and enhanced DNA damage with CX-3543 treatment in the ATRX-deficient context. These findings are presented in FIG. 1d-1e and Supplementary FIG. 7, are described in the Results section, pp. 5-6 and 9, and briefly mentioned in the Discussion section, pp. 13 and 14.

2- Also, the authors should comment if the effects they see in the NHA cells ATRX deficient reported in their manuscript would hold for all ATRX deficient glioma cell types independently of other co-expressed genetic lesions; or if the presence of other genetic alterations would have an impact on the phenotype they describe.

We thank the reviewer for this insightful suggestion. In adult gliomas, ATRX deficiency almost invariably co-occurs with both p53 deficiency (usually through mutation of the TP53 gene) and IDH mutation (involving either IDH1 or IDH2). It should be noted that our primary cell line model in this study, the NHA, is p53-deficient due to E6/E7 overexpression. We also now include ATRX-intact and ATRX-deficient NHAs harboring the glioma-associated IDH1 R132H mutation. We find that the addition of IDH1 mutation has no appreciable effect on G4 formation above and beyond what is associated with ATRX deficiency. These data are now presented in Supplementary FIG. 1 and described in the Results section, p. 5. Moreover, our studies in response to comment #1 (see above) utilizing TS 603 and 08-0537 GSCs, both of which feature IDH1 mutation, are also relevant here, in that they completely recapitulate the relevant NHA findings.

3- The rationale for the comparisons shown in Figure 3 need to be better justified/explained by the authors in the context of molecular glioma subtypes.

We have reworded this section (Results, pp. 7-8) to better emphasize the relevance of these findings to the IDHmut-noncode1 glioma subtype, which is defined by ATRX deficiency in adult populations.

4- Data reported in Figure 5 should be repeated using the patient derived human glioma cells which are ATRX deficient.

We appreciate this suggestion. We have now repeated these studies with ATRX-intact and ATRX-knockdown TS 543 GSCs, finding similar results. These data are presented in Supplementary FIG. 10 and described in the Results section, p. 10.

5- The authors could include a diagram related to the proposed mechanism of action of the therapeutic intervention in ATRX mutant gliomas.

Thank you for this suggestion. We now include this diagram in FIG. 9, and refer to it in the Discussion section, p. 14.

6- The authors should also revise the discussion in relation to the impact of IDH1 mutations on DNA repair (refs 53-55), as none of the papers cited, assess the impact of IDH1 mutation in the context of ATRX mutations, which is the main topic addressed in this manuscript. One of the cited papers is related to AML. the other paper used colon cancer cells and Hela transfected with mutant IDH1 and the third paper uses U87 and U251 cells transfected with mutant IDH1, none of these are relevant to the current manuscript nor to any relevant patient derived glioma cell culture expressing mIDH1 and harboring ATRX deficiency.

We appreciate this comment. The relevant sentence and cited references have been removed from the Discussion section, p. 15.

Very sorry again to bother you with this. I have followed your recent work with great interest. It has definitely inspired/informed a major line of investigation for my lab.

Yours truly,

Jason

Jason T. Huse, MD, PhD
Associate Professor
Departments of Pathology and Translational Molecular Pathology
University of Texas MD Anderson Cancer Center
phone:713-745-3186
Email: jhuse@mdanderson.org

We would like to thank the reviewers, once again, for their insightful commentary. Please find below, **in bold**, our point-by-point response to specific reviewer comments.

Reviewer #1 (Remarks to the Author):

I will limit my comments to how my reviewer comments have been addressed. This manuscript is greatly improved by the additional experiments that bolster the authors' argument that G4 structures are responsible for genomic instability in ATRX deficient neuronal cells and that this may be exploited through the use of G4 stabilisers. They have confirmed the findings associated with the antibody IH6 with an independent antibody BG4. I'm rather disappointed they did not confirm their hf2 ChIP findings with BG4, which has been used for ChIP - given the paucity of publications using these reagents I would have been happier to see confirmation with more than one antibody in any one experiment.

We appreciate that our revisions have been generally well received. While we did not specifically include a BG4 ChIP experiment, we feel that our BLM ChIP studies serve an equivalent purpose, in that they establish genomic correlations between DNA damage and established G4 binding proteins.

It is gratifying to see them using BLM antibodies to strengthen the argument that the DNA damage observed in ATRX deficient cells is at sites of G4. Again this greatly enhances their argument.

Thank you.

One thing that should be addressed is the lack of quantitation of the IF results in particular for Suppl Fig 1c, Suppl Fig 1d and Suppl Fig 8b - for example they describe the results in Suppl Fig 1d as 'significant' without demonstrating that this is true.

We appreciate this comment. We have now provided quantification of G4 IF signal for Supplementary FIG. 1c, 1d, and 8b. We have also included correlation coefficients for G4 and 53BP1 signal overlap in Supplementary FIG. 8b.

The validation of ATRX knockdown by sh590 (shown in Fig 5d) should be referred to at an earlier stage - when first used on page 5.

Thank you for this suggestion. The ATRX western blot showing sh590-mediated knockdown in TS 543 cells is now referred to on p. 6 and shown in Supplementary FIG. 1e.

Reviewer #2 (Remarks to the Author):

This manuscript relates to mechanisms elicited by mutational inactivation of ATRX (α -thalassemia mental retardation X-linked), present in large subsets of malignant glioma. The

authors aimed to uncover the pathogenic consequences of ATRX deficiency, both in vitro and in vivo. In this manuscript, the authors report that ATRX loss in isogenic glioma model systems induces replication stress and DNA damage by way of G-quadruplex (G4) DNA secondary structure. Data is presented to support that these effects are associated with the acquisition of copy number alterations over time. The data presented also demonstrates both in vitro and in vivo, that ATRX deficiency leads to increased DNA damage and cell death following chemical G4 stabilization. The authors also report that G4 stabilization in an ATRX-deficient context, synergizes with other DNA-damaging therapies, such as ionizing radiation.

The experiments are well designed and rigorous.

The authors have applied adequate statistical analysis of their data. The conclusions are in line with the results presented by the authors.

The authors also need to be commended for having performed extensive further experimentation to address all the comments raised during the original review of their manuscript. This work is both novel and of high impact to the field.

We greatly appreciate this very positive feedback. Thank you!